# Fully Depleted, Trench-Pinned Photo Gate for CMOS Image Sensor Applications

**DOI:** 10.3390/s20030727

**Published:** 2020-01-28

**Authors:** Francois Roy, Andrej Suler, Thomas Dalleau, Romain Duru, Daniel Benoit, Jihane Arnaud, Yvon Cazaux, Catherine Chaton, Laurent Montes, Panagiota Morfouli, Guo-Neng Lu

**Affiliations:** 1STMicroelectronics, 850 Rue Jean Monnet, 38921 Colles, France; andrej.suler@st.com (A.S.); thomas.dalleau@st.com (T.D.); romain.duru@st.com (R.D.); daniel.benoit@st.com (D.B.); jihane.arnaud@st.com (J.A.); 2IMEP-LaHC, Université Grenoble Alpes, 38016 Grenoble, France; Laurent.Montes@grenoble-inp.fr (L.M.); panagiota.morfouli@phelma.grenoble-inp.fr (P.M.); 3INL, Université Claude Bernard Lyon 1, 69622 Villeurbanne, France; guo-neng.lu@univ-lyon1.fr; 4LETI-CEA Tech, 17 rue des Martyrs, 38054 Grenoble, France; yvon.cazaux@cea.fr (Y.C.); catherine.chaton-ext@st.com (C.C.)

**Keywords:** CMOS image sensor (CIS), pixel, photo gate, transfer gate, capacitive deep trench isolation, surface passivation, dark current

## Abstract

Tackling issues of implantation-caused defects and contamination, this paper presents a new complementary metal–oxide–semiconductor (CMOS) image sensor (CIS) pixel design concept based on a native epitaxial layer for photon detection, charge storage, and charge transfer to the sensing node. To prove this concept, a backside illumination (BSI), p-type, 2-µm-pitch pixel was designed. It integrates a vertical pinned photo gate (PPG), a buried vertical transfer gate (TG), sidewall capacitive deep trench isolation (CDTI), and backside oxide–nitride–oxide (ONO) stack. The designed pixel was fabricated with variations of key parameters for optimization. Testing results showed the following achievements: 13,000 h+ full-well capacity with no lag for charge transfer, 80% quantum efficiency (QE) at 550-nm wavelength, 5 h+/s dark current at 60 °C, 2 h+ temporal noise floor, and 75 dB dynamic range. In comparison with conventional pixel design, the proposed concept could improve CIS performance.

## 1. Introduction

Conventional pixel design for complementary metal–oxide–semiconductor (CMOS) image sensors (CIS) commonly uses fully depleted photodiodes as a photo-sensing element. With rapid CIS developments, the photodiode structure evolved from the early planar pinned photodiode (PPD) [1,2] to the current deep PPD [3], proposed to address the backside illuminated (BSI), high-resolution image sensor market. Even so, this technology is limited by the need to employ ion implantation that can cause crystal damage and metal contamination [4,5]. Indeed, to meet high-resolution imaging requirements, shrinking the pixel size led to space search in silicon depth for the storage of photo-generated charges. As charges are stored in deep PPD, it becomes more difficult to drain them via a surface transfer gate (TG) for the reading. To help vertical charge draining toward the surface TG, three-dimensional (3D) gradual doping through careful implantation control can create an electric field. Moreover, high-energy ion implant is necessary for the definition of deep PPD. However, such implantation processes cause defects that may not be entirely suppressed due to limited annealing temperatures, and there are risks of introducing undesirable impurities. This induces deep-level traps in silicon, leading to dark current degradation.

We propose in this paper a novel concept for pixel implementation, introducing a vertical pinned photogate (PPG) to replace deep PPD using epitaxial silicon as the active layer. In continuation of the former planar charge-coupled device (CCD) and implanted trench photo MOS concept [6,7,8,9], the proposed integration requires no implantation [10], thus avoiding issues of defect creation and contamination. Secondly, to efficiently drain charges stored in the PPG, the development of a buried vertical TG in pipe form between the charge storage region and surface sensing node (SN, also called floating diffusion or FD) is proposed. This makes the transfer path direct in comparison with the conventional implementation of gradual doping and surface TG. Moreover, the proposed pixel structure takes benefit from capacitive deep trench isolation (CDTI), formerly developed for dark current reduction [11] and more recently for fully depleted memories for global shutter applications [7,12]. Its use as a pixel sidewall allows active lateral surface passivation with surface potential pinning for the PPG. Finally, for backside surface passivation, an electrostatic approach by integration of an oxide–nitride–oxide (ONO) stack is proposed. The ONO stack holds a certain quantity of charges to establish, by field effect, a sufficient density of oppositely charged carriers at the silicon (backside) surface. It also plays the role of a photon transmission layer through its optical low-absorption and antireflection properties in the visible spectrum.

To assess the proposed concept, a BSI, p-type, 2-µm-pitch pixel (in square layout) with vertical TG (in square layout) was designed (see illustration in Figure 1a) to operate at a supply voltage range of [0 V, 3.3 V], using 90-nm-node CMOS facilities. The pixel was fabricated (shown in Figure 1b) in test structures with variations of key parameters for optimizing performance. The achieved performance was evaluated by measuring the test structures. 

## 2. Pixel Description

Figure 1a illustrates the designed BSI pixel with integration of a vertical PPG, and a buried vertical TG beneath a surface SN, a sidewall CDTI, front-side readout transistors, and a back-side ONO stack. The pixel structure is implemented from an epitaxial silicon layer of a few micrometers.

### 2.1. Vertical Pinned Photo Gate (PPG)

A commonly used material for the digital CMOS core process is an epitaxial p-type substrate, called the vertical PPG, which was implemented in this epitaxial silicon layer with a uniformly doped concentration N_A_. This substrate specification implies that the signal charges are holes to be collected and stored in the PPG. The silicon volume of the pixel is limited by sidewall CDTI, and the PPG takes a large part of it with a good fill factor in volume. By surface passivation of sidewall CDTI, the PPG forms a potential well (shown in Figure 2a,b) to collect and store photo-generated holes. The potential well can be determined by solving the Poisson equation [1]. Issued from a one-dimensional solution, the *X*-axis (horizontal, passing through the PPG’s center) potential profile is given by
*V*(*x*) = (*qN_A_*/2 *ε_Si_*)(*x* − *W_d_*/2)^2^ + *V_D_*.(1)

From the obtained profile, the minimum potential at the PPG’s center, called depletion voltage *V_D_*, and the potential well’s depth, ∆V_D_, can be extracted. They have the following relationship:*V_D_* = *V_S_* + ∆*V_D_*,(2a)
with
∆*V_D_* = −(*W_d_*^2^/8)(*qN_A_*/*ε_Si_*),(2b)
where *V_S_* is the lateral surface potential, and *W_d_* is the silicon width between face-to-face CDTI gate electrode. *V_S_* is pinned when the lateral Si–SiO_2_ interface is set in inversion mode via CDTI bias control. Such an operation mode cancels the thermal generation of dark current from this interface [11,13].

The parameters involved in relationship (2b) are the key to pixel operation with optimal performance because of the tradeoff between full depletion voltage, transfer efficiency, and full well. In particular, increasing ∆*V_D_* improves full well capacity, but also lowers *V_D_*, thus reducing the potential difference (versus SN) for charge transfer.

For design optimization and determination of key parameters, two-dimensional (2D) and 3D process and device simulations were carried out.

### 2.2. Vertical Transfer Gate

There was a preliminary investigation into lateral deep transfer gates [14]. However, for this pixel structure, we preferred a vertical shallow trench transfer gate, placed in the central surface area of the pixel (see Figure 3a), in front of the maximum fully depleted photo gate voltage position to maximize the charge transfer efficiency at low operating voltage.

The vertical TG in pipe form surrounds a wire-shaped channel for charge transfer from PPG to surface SN. It needs to ensure both charge holding and transfer operations. In the off state, a potential barrier for stored holes as high as ∆V_D_ is desirable, while, in the on state, the *Y*-axis (vertical, passing through the PPG’s center) potential profile from the backside surface of the PPG to the front-side surface SN should be in a monotonic decrease for holes, so as to completely transfer the stored charges without lag. Figure 3b depicts the vertical potential profile for different TG voltages.

The vertical TG is buried with its controlling electrode below the silicon surface, thus leaving its surface area to SN. This allows SN to be implemented with smaller capacitance, so as to enhance the charge-to-voltage factor (CVF).

### 2.3. Front-Side Readout Transistors

For the pixel readout, the conventional 2T5 pixel architecture was adopted with the use of PMOS transistors. This architecture is compatible with the correlated double sampling (CDS) readout mode. The pixel makes use of front-side planar readout transistors implemented in a shallow n-type well surrounding the buried TG (shown in Figure 1a,b). The well region also works as a top pinning layer of PPG, and as a source of carriers (electrons) for the lateral and bottom surfaces to reach the inversion mode.

### 2.4. Backside Positively Charged ONO Stack

The ONO stack can be integrated by successive deposition of three dielectric layers on the backside silicon surface, using plasma-enhanced chemical vapor deposition (PECVD). The first deposited layer, oxide, plays the following roles: (1) chemical surface passivation, with lower interface state density at the Si–SiO_2_ interface than the case of nitride deposition on silicon [15]; (2) retaining charges of the second deposited layer, with reproducible parameters. The second layer is hydrogenated silicon nitride SiN_x:H, and the thin film can be positively charged thanks to its K centers [16], thereby playing the role of a field-effect passivation and antireflection layer. The third layer is oxide, deposited mainly to complete the optical antireflective effect. 

With antireflective consideration for a nearly 100% light transmission ratio at 550-nm wavelength, the three stacked layers were chosen to have the following thicknesses: 20 nm for the first oxide, 60 nm for the nitride, and 170 nm for the last oxide.

Based on Sentaurus’s electrical simulations (see Figure 4), the ONO stack needs to hold a surface density of positive charges over 5 × 10^11^ cm^−2^ at the Si–SiO_2_ interface to ensure backside surface passivation with the surface potential pinned to the value imposed by the n-type well of PMOS transistors.

Experimentally, the ONO stack was deposited onto silicon wafers in different configurations, and the COCOS characterization technique (corona oxidation for characterization of semiconductors) was employed to estimate the Si–SiO_2_ interface states and fixed charge distribution [17]. Those dielectric parameters were correlated to dark current measurements [15]. The best configuration possessed a density of fixed charges of 6.9 × 10^11^ cm^−2^ and a density of interface states of 4.5 × 10^9^ eV^−1^∙cm^−2^. It was chosen and integrated in the pixel fabrication process.

## 3. Results and Discussion

The designed p-type, 2-µm-pitch pixel was fabricated using 90-nm MOS technology with wafer backside thinning and ONO stack deposition. It was integrated in test structures with variations of key parameters such as the epitaxial layer thickness (from 2.7 to 4.8 µm) and doping concentration *N_A_* (two levels: A and B). The test structures were measured to evaluate the pixel characteristics, including full well capacity, quantum efficiency (QE), and dark current.

### 3.1. Full Well Capacity and QE

The pixel under test operated properly with no lag observation during charge transfer. Figure 5a plots the full well capacity against epitaxial layer thickness for two doping levels A and B. One can observe a 1.7-µm equivalent zone without any charge storage. This zone comes from top and bottom space charge extension and vertical transfer gate depth. Beyond this zone, the full well capacity went up linearly with the epitaxial layer thickness. In addition, it went up more quickly for a higher doping level B, reaching 13,000 h+ for a 4.8-µm silicon thickness. 

Figure 5b represents QE as a function of wavelength for three values of epitaxial layer thickness. QE was improved when increasing the thickness, which can be explained by the fact that PPG had more silicon space (in depletion) to absorb incoming photons and collect photo-generated holes. For the 4.8-µm epi layer thickness, the maximum QE reached 80% at 550 nm (pixel without color filter and micro lens). It is noted that QE for large wavelengths was enhanced mainly because there were reflecting layers and components on the front side of the pixel, which roughly doubled the silicon absorption thickness.

### 3.2. Dark Current

Dark current of the p-type pixel was measured and compared with an n-type counterpart: 1.75-µm-pitch, planar pinned photodiode, BSI, with CDTI pixel isolation.

Figure 6a compares statistic distributions of dark current measured at 60 °C from the test pixel array. There were two types of pixels: p-type PPG and n-type PPD. The p-type exhibited the intrinsic peak (representing normally operated pixels) at 4.5 h+/s with a much smaller dispersed population. In contrast, the n-type counterpart peaked at 17e^−^/s with a larger tail, resulting in an average value of 23 e^−^/s. The large tail distribution of the n-type may be due to high-energy implantation process steps needed for its fabrication. 

Measuring the dark current’s temperature dependence permits extraction of the activation energy, which indicates the predominant mechanism of dark current generation [18]. We performed dark current measurements on the p-type pixel array at and above 60 °C, below which signals were too weak to be practically measurable. The intrinsic pixels with dark current centered at 4.5 h+/s had their activation energies around 1.1 eV, corresponding to the silicon bandgap. This means that dark currents in the intrinsic pixels were dominated by a diffusion mechanism. It also means that the silicon surface surrounding the PPG was effectively passivated. The remaining pixels of higher values of dark current (peaks at around 100 h+/s and above) had their extracted *E_a_* near 0.8 eV.

Figure 6b compares standard deviations of dark currents between the two types of pixel arrays, i.e., p-type and n-type. The differences were significant, e.g., at 100 °C, the value was ~200 h+/s for the p-type compared with ~1000 e^−^/s for the n-type. The p-type with smaller standard deviation denoted that the pixel structure was more robust to temperature elevation.

### 3.3. Main Characteristics

Other measured results included a conversion voltage factor (CVF) of 90 μV/h+, a photo-response non-uniformity (PRNU) of 0.5%, and a temporal noise floor of 2 h+. Thanks to the high full well capacity for the thick structure, a dynamic range of 75 dB was estimated. Table 1 summarizes the main performances of the designed pixel.

## 4. Conclusions

We designed and realized a 2T5 rolling shutter image sensor, p-type, and 2-µm-pitch pixel to evaluate and validate our proposed concept based on a native epitaxial layer zone for photon detection, charge storage, and charge transfer to the sensing node. It integrates a vertical PPG, buried vertical TG, sidewall CDTI, and backside ONO stack. Testing results show a high QE, high full well capacity with no lag, and low dark current. In comparison with the conventional pixel design, the proposed concept can bring better performance. This new structure featuring a silicon trench etching process may be deployed for either large or small pixel sizes, as well as thin or deep silicon thicknesses, targeting a photo response ranging from the ultraviolet (UV) to near-infrared (NIR) wavelength.

## Figures and Tables

**Figure 1 sensors-20-00727-f001:**
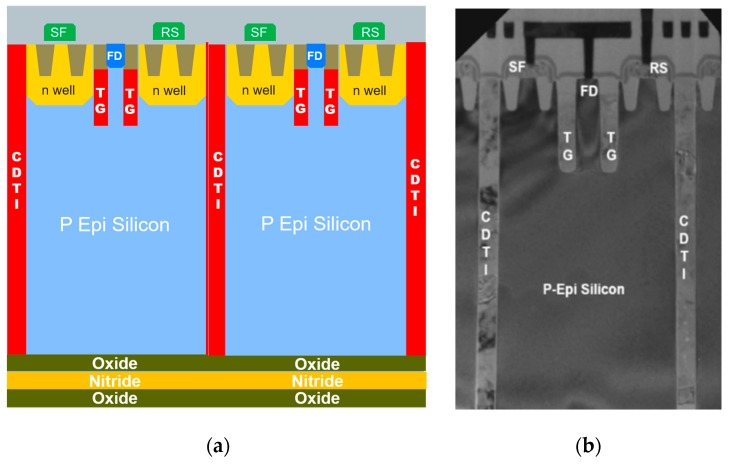
(**a**) Pixel structure integrating vertical PPG, vertical TG, sidewall CDTI, front-side planar readout PMOS transistors, and backside ONO stack; (**b**) TEM cross-section image of the fabricated p-type, 2-µm-pitch pixel.

**Figure 2 sensors-20-00727-f002:**
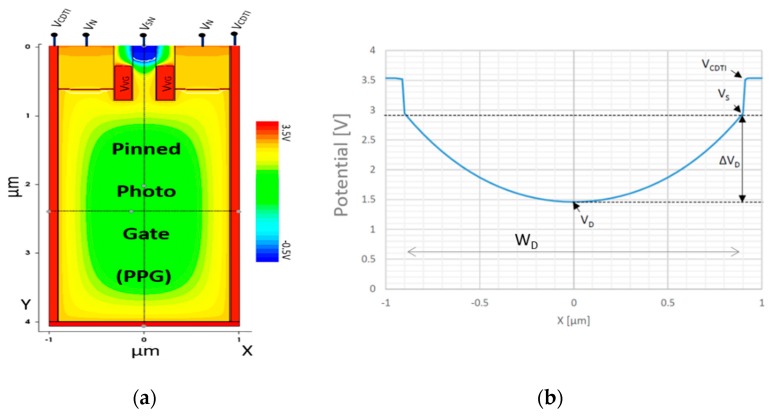
(**a**) Device schematic with electrostatic potential distribution, and (**b**) electrostatic potential diagram cross-section at *y* = 2.5 µm of the proposed architecture.

**Figure 3 sensors-20-00727-f003:**
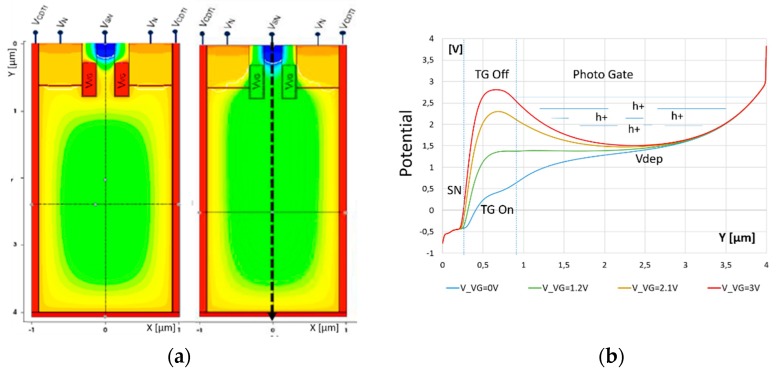
(**a**) Electrostatic potential diagram, TG off (left), TG on (right), (**b**) vertical, *Y*-axis, electrostatic potential profile at *x* = 0 µm for TG switched from off to on (3 V, 2.1 V, 1.2 V, 0 V).

**Figure 4 sensors-20-00727-f004:**
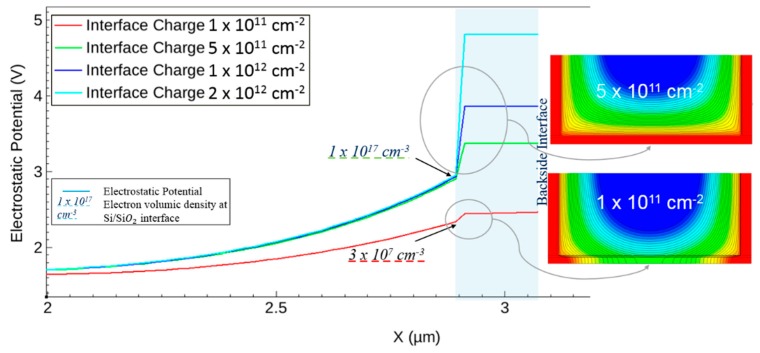
Electrostatic simulation of backside interface with electrostatic potential (**left**) and the corresponding potential (**right**).

**Figure 5 sensors-20-00727-f005:**
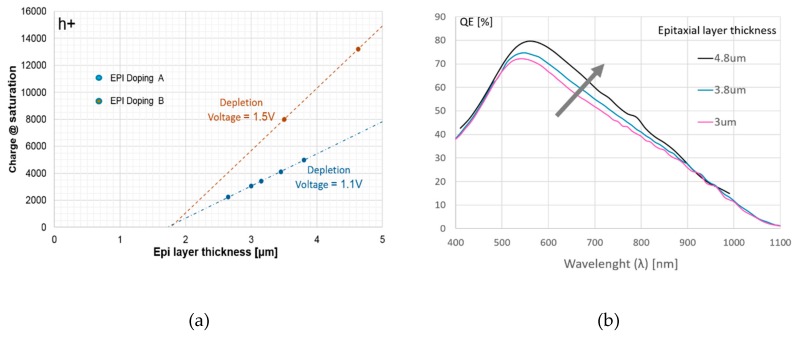
(**a**) Full well capacity versus epitaxial layer thickness for two doping levels (A and B); (**b**) quantum efficiency (QE) of the pixel (without color filter and micro lens) versus wavelength (*λ*).

**Figure 6 sensors-20-00727-f006:**
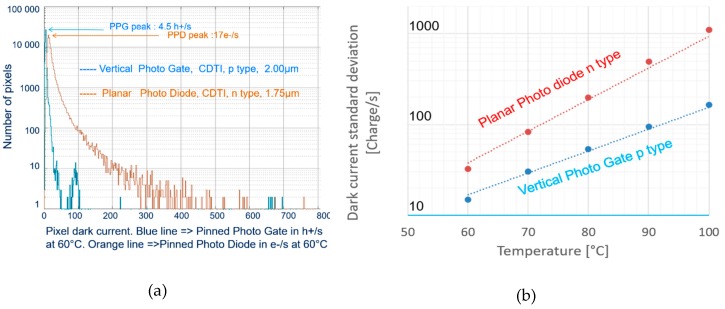
(**a**) Dark current statistic distribution of the p-type and n-type pixel array, from measurements at 60 °C; (**b**) dark current standard deviations of the p-type and n-type pixel array versus temperature.

**Table 1 sensors-20-00727-t001:** Pixel performances.

Characteristics	
Pixel	P type, 2.0 µm pitch, 2T5, RS, 103,000 pixels
CVF	90 µV/h+
Saturation charge	13,000 h+
QE at 550nm	80%
PRNU	0.5%
Mean dark current at 60 °C	5.3 h+/s
Lag	< 1 h+
Temporal noise floor	2 h+
Dynamic range	75 dB

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
