# Peer review of "Fully Depleted, Trench-Pinned Photo Gate for CMOS Image Sensor Applications"

_sensors, 2020, doi:10.3390/s20030727_

Round 1

Reviewer 1 Report

This work is an expanded version of a proceeding paper presented at the 2019 International Image Sensor Workshop (IISW), Snowbird, Utah, USA, June 23-27, 2019 and published online:

Roy et al. “Side Illuminated, Fully Depleted, Pinned Trench Photo MOS for Imaging Applications”

Although many of the results and figures are in common with this paper, I think that it is worth considering a full paper, where the concept can be expressed much more in detail, for publication.

The original proceeding paper, however, should be cited and it should be paid attention to copyright issues if the same figures/graphs are used.

The article is clearly written. I have only a few comments.

Minor: Page 2, rows 75-78: the symbol vs should be written as Vs (uppercase V) to match the notation used in Fig. 1 b.

Major: The QE in Figure 6b above 900nm appears unrealistically high for the considered thickness. Since the absorption length at 900nm is of the order of 30micrometers, a QE of almost 30% for a layer of 3-5 microns appears too large. In addition, QE should be dependent on the oxide thickness also in this spectral region, while in figure 6b it is not. Authors should justify this behavior or fix the measurement and include a correct plot.

Minor: labels are not readable in Figure 7a. In general, the graphs appear as bitmap images with low resolution. The resolution of the graphs should be improved.

Minor: in page 6, line 176, it is stated that an activation energy of 1.14eV represents the intrinsic dark current contribution. This deserves a better explanation, and a reference should better be provided. In photodiodes and CCDs, SRH generation in the depletion region gives an activation energy of EG/2, while the injection in the depletion region of carriers diffusing from neutral regions gives an activation energy of EG. See for example [Ralf Widenhorn et al., “Temperature dependence of dark current in a CCD”, Proceedings of SPIE Vol. 4669]. In this case, in most of the pixels the SRH generation is negligible, so are the generated holes diffusing from the neutral surface nwell region? Is the same value of Ea observed also at temperatures lower than 60C?

Author Response

Reviewer 1 :

The article is clearly written. I have only a few comments.

Minor: Page 2, rows 75-78: the symbol vs should be written as Vs (uppercase V) to match the notation used in Fig. 1 b.

Response : Done.

Major: The QE in Figure 6b above 900nm appears unrealistically high for the considered thickness. Since the absorption length at 900nm is of the order of 30micrometers, a QE of almost 30% for a layer of 3-5 microns appears too large. In addition, QE should be dependent on the oxide thickness also in this spectral region, while in figure 6b it is not. Authors should justify this behavior or fix the measurement and include a correct plot.

Response : The QE above 900nm looks unrealistically high, but it is confirmed by both measurements and simulations.  We have added the following sentences before the figure for explaining this :

It is noted that QE for large wavelengths is enhanced mainly because there are reflecting layers and components on the front side of the pixel, which roughly doubles the silicon absorption thickness.

Minor: labels are not readable in Figure 7a. In general, the graphs appear as bitmap images with low resolution. The resolution of the graphs should be improved.

Response : Improvements on figrure presentation have been done.

Minor: in page 6, line 176, it is stated that an activation energy of 1.14eV represents the intrinsic dark current contribution. This deserves a better explanation, and a reference should better be provided. In photodiodes and CCDs, SRH generation in the depletion region gives an activation energy of EG/2, while the injection in the depletion region of carriers diffusing from neutral regions gives an activation energy of EG. See for example [Ralf Widenhorn et al., “Temperature dependence of dark current in a CCD”, Proceedings of SPIE Vol. 4669]. In this case, in most of the pixels the SRH generation is negligible, so are the generated holes diffusing from the neutral surface nwell region? Is the same value of Ea observed also at temperatures lower than 60C?

Response : The suggested reference has been added and cited ([18]).

Reviewer 2 Report

In general nice paper specially in view of the fact is uses holes whereas the majority are on electrons.   The use of PPG is somewhat misleading because it no gate it is just a slab of silicon. So when one looks in the figures one does not find a gate in the normal sense. 
Maybe a few words on why the use of gate in PPG.

When printed the numbers and abbreviation in the figures come out weak. Use bold or larger sized characters/numbers.
  p1:
line 15: typo, Back Size=>Back Side
  p2:
line 44: TG in pipe form.
The top view of the pipe you made is that circular or square? And the pixel is it a square or a circular one?
line 60: for readability add reference to figure 3 too.
line 67: silicon space=> silicon volume.
  p4:
Figure 3 is a much more illustrative, if possible move it to the beginning of 2. Pixel description.
  p5:
line 138: can you define the best configuration in numbers of interface states and fixed charges? Or can you quantify it?

p6:
Figure 7a the orange color is difficult to read when printed. Idem for the grid of the graphs.

Caption 7b uses "intrinsic dark current" the graph Y-axis "Dark std-dev" which is it?
line 173 and Figure 7b: what is the definition of "dark standard deviation"?  Do you mean FPN, or Shotnoise of darkcurrent, or Darkcurrent?
line 180 definition of current deviation? idem
Looking at the use of e/sec and the caption of fig 7b you probably mean darkcurrent. But in fig 7b the Eact is about 0.6eV and you state 1.14eV so then it is the sqrt of the dark current or shotnoise. But then....Line 175: 5 h+/s at 60C but the figure 7b stdev is 15 h+/sec???????
THINGS DO NOT ADD UP. CORRECT SVP.
Figure 7b: To make sure: how did you determine the "dark standard deviation" at what integration time?

p6:
Table 1:
add the number of pixels of the device on which the measurements are performed on.
p7:
Table 1:
Lag: instead of "no lag" give the lowest value you can measure.

Author Response

Reviewer 2 :

In general nice paper specially in view of the fact is uses holes whereas the majority are on electrons.   The use of PPG is somewhat misleading because it no gate it is just a slab of silicon. So when one looks in the figures one does not find a gate in the normal sense. 
Maybe a few words on why the use of gate in PPG.

Response : The device structure has been called as pinned photo gate (PPG) in our published papers (Ref. 10, etc.). So we continue to use PPG for being coherent.

When printed the numbers and abbreviation in the figures come out weak. Use bold or larger sized characters/numbers.   p1:

Response : Done.

line 15: typo, Back Size=>Back Side   p2:

Response : Correction done.

line 44: TG in pipe form. The top view of the pipe you made is that circular or square? And the pixel is it a square or a circular one?

Response : Done with square indication for the pixel and the TG (lines 61-62).

line 60: for readability add reference to figure 3 too.

Response : Done with figure 3 moved as figure 1a.

line 67: silicon space=> silicon volume.   p4:

Response : Done.

Figure 3 is a much more illustrative, if possible move it to the beginning of 2. Pixel description.   p5:

Response : Done with figure 3 moved as figure 1a.

line 138: can you define the best configuration in numbers of interface states and fixed charges? Or can you quantify it?

Response : Done with added densities of interface states and fixed charges (in lines 151-152).

p6:
Figure 7a the orange color is difficult to read when printed. Idem for the grid of the graphs.
Caption 7b uses "intrinsic dark current" the graph Y-axis "Dark std-dev" which is it?

Response : Improvements of the two figures (Figure 6a and 6b for the revised version) have been made.

line 173 and Figure 7b: what is the definition of "dark standard deviation"?  Do you mean FPN, or Shotnoise of darkcurrent, or Darkcurrent?

Response : A statistical distribution of pixels’ dark currents can be characterized by standard deviation for its dispersion. We have Changed textual description in lines 198-201 to improve clarity.

line 180 definition of current deviation? Idem

Response : We mean standard deviation. Correction done.

Looking at the use of e/sec and the caption of fig 7b you probably mean darkcurrent. But in fig 7b the Eact is about 0.6eV and you state 1.14eV so then it is the sqrt of the dark current or shotnoise. But then....Line 175: 5 h+/s at 60C but the figure 7b stdev is 15 h+/sec???????
THINGS DO NOT ADD UP. CORRECT SVP. Figure 7b: To make sure: how did you determine the "dark standard deviation" at what integration time?

Response : The presented results in fig 7b (fig. 6b in the revised version) was mistakenly presented for extracting activation energy. Correction done with description changes in lines 190-197.

p6:
Table 1:
add the number of pixels of the device on which the measurements are performed on. p7:

Response : Done

Table 1:
Lag: instead of "no lag" give the lowest value you can measure.

Response : Correction done with Lag < 1h+